# Characteristics of Mortar Containing Oyster Shell as Fine Aggregate

**DOI:** 10.3390/ma15207301

**Published:** 2022-10-19

**Authors:** Ui-In Jung, Bong-Joo Kim

**Affiliations:** 1Research Center for Environment Friendly Concrete, Department of Architectural Engineering, Greensmart, KongJu National University, Cheonan 31080, Korea; 2Department of Architectural Engineering, Greensmart, KongJu National University, Cheonan 31080, Korea

**Keywords:** oyster shell, aggregate, mortar, fire resistance, thermogravimetric analysis, porosity, fire-extinguishing effect, endothermic reaction

## Abstract

In this study, oyster shells were processed and classified into sizes equal to or smaller than the fine aggregate threshold, and their engineering properties and fire-resistant performance were examined. The differences in heating weight loss of oyster shell aggregate (OSAs) with different particle sizes were examined using thermogravimetric analysis (TGA). The TGA results showed indicating that the temperature at which decarboxylation reaction started depended on the OSA particle size. The porosity of mortar specimens was analyzed using mercury intrusion porosimetry (MIP). The porosity area and porosity of the OSA-containing mortar increased with decreasing particle size. Mortar fire-resistant boards with heated for 2 h in accordance with the heating conditions of KS F 2257-1(methods of fire-resistant testing for structural element—general requirements) to measure their back-side temperature. The board made with OSA2.5 exhibited 273.2 °C, which is more than 90 °C higher than the back-side temperature of the board with OSA 0.6Under. Such difference was attributed to the greater heat transfer delay caused by higher porosity, porosity area, and specific surface area in OSAs with small particle sizes. The TGA results combined with the heating test results suggested that CO_2_ would be generated at different temperatures in boards containing OSAs with different particle sizes because of the differences in the endothermic reaction temperature.

## 1. Introduction

Fire-resistant finishings of buildings are mainly classified into wet and dry types. In general, the wet type is constructed with fire-resistant materials, such as fire-resistant paint and spray coating, and the dry type is constructed using fire-resistant boards [1].

For wet construction, an additional curing period is required after construction, and the final durability depends on the quality of construction. In addition, repair is required over time because of scaling and spalling. In cases of dry construction, a separate curing period is not required, and precise construction is possible [1,2,3]. However, for both wet and dry types, the damage to the structure is hard to prevent in case of fires accompanied by explosions and impacts exceeding a certain level because these materials do not possess sufficient durability. Therefore, fire-resistant boards with high strength that can withstand explosions and impacts are required [4,5].

Oyster shells are mainly composed of CaCO_3_ (approximately 96%) and are used as a filler in the cement matrix without a negative effect on cement hydration [6]. Various studies have investigated the recycling of oyster shells. According to the results of studies on the basic properties of oyster shells as a construction material, their strength and permeability are similar to those of sand, which is a typical fine aggregate; additionally, oyster shells are lightweight and easy to crush [6,7,8].

Oyster shells are industrial waste generated from oyster farming. They are difficult to recycle; therefore, oyster shells accumulate in large quantities on the coast close to the farms. Over time, organic matter left on the illegally disposed oyster shells decays, ultimately causing environmental problems along the coast and leading to the destruction of coastal ecosystems owing to the foul decay odor, inhibition of plankton growth, and solidification of tidal flats [6,9,10,11].

Oyster shells have been confirmed to delay heat transfer and inhibit fire because of the endothermic decomposition of calcium carbonate at 800 °C. In addition, as shown in the following equation, calcium carbonate decomposes into calcium oxide and carbon dioxide, which has fire-extinguishing properties [5,12].
$$CaCO_{3}\vec{heat\left(energy\right)}CaO+CO_{2}$$

The performance of fire-resistant boards containing oyster shells as fine aggregate particles has already been investigated. The back-side temperatures of the oyster shell-containing boards have been shown to be lower than those of conventional mortar boards under all particle size conditions. At the same time, the measured back-side temperature, considered a standard for fire-resistant performance, was lower for mortar boards with oyster shell particles smaller than 0.6 mm than for those with particle sizes between 2.5 and 5.0 mm. Further research and analysis, however, are required to investigate the reasons and causes of the differences in fire-resistant performance for different particle sizes. Therefore, a method for selecting the appropriate particle size range for the required fire-resistant performance needs to be developed [5,6,12].

Therefore, in this study, oyster shells were processed and classified into sizes equal to or smaller than the fine aggregate threshold, and their engineering properties and fire-resistant performance were examined to provide data for utilizing oyster shells as an aggregate for fire-resistant boards.

## 2. Experimental Plan

### 2.1. Experimental Scope and Method

In this study, the engineering properties of the oyster shell aggregate (OSA), which was classified according to the particle size, and OSA-containing mortar were examined. Particularly, the density, water absorption, unit volume weight, and solid volume percentage of OSA were measured. The experiment was performed in accordance with KS F 2504 (test method for density and absorption of fine aggregates) and KS F 2505 (test method for unit volume weight and solid volume percentage of aggregates) [13,14]. The differences in heating weight loss of OSAs with different particle sizes were examined using thermogravimetric analysis (TGA). The crystal structures of the theoretically known components of cement and oyster shells were examined, compared, and analyzed. Subsequently, mortar specimens with OSA with different particle sizes were prepared, and their properties were examined. They were prepared at the flow range of 180 ± 5 mm and tested according to KS L ISO 679 (methods of testing cements—determination of strength) [15]. The porosity of mortar specimens was analyzed using mercury intrusion porosimetry (MIP). Mortar fire-resistant boards with a size of 300 × 300 × 50 mm (width × length × thickness) were fabricated, installed on a wall, and heated for 2 h in accordance with the heating conditions of KS F 2257-1 (methods of fire-resistant testing for structural element-general requirements) to measure their back-side temperature, as shown in Figure 1 [16].

### 2.2. Materials

The materials used in this study were oyster shells and ordinary Portland cement, and their properties are detailed subsequently.

#### 2.2.1. Oyster Shells

Oyster shells used in this study were subjected to primary grinding at an oyster processing site in Tongyeong, Korea. The recovered oyster shells were washed, dried, and ground in a cutter mill, and then classified according to the particle size using a screen (Figure 2).

Oyster shells were used in the form of fine aggregate particles (5 mm or less) because the flat and elongated shape of oyster shells may lower the workability of the mortar and cause differences in strength owing to the formation of unintended internal pores in the regions where oyster shells overlap after mixing. The above process was also performed for greater precision, so the organic matter on the oyster shells was removed. As shown in Figure 3 and Figure 4, OSA obtained after grinding was classified into fractions with particle sizes of 0.6 mm or less (OSA0.6U), 0.6–1.2 mm (OSA0.6), 1.2–2.5 (OSA1.2), and 2.5–5.0 mm (OSA2.5) to examine the dependence of properties on the particle size [17]. The chemical composition of the oyster shells is listed in Table 1. Similar to previous studies, CaO comprised 53.66%, and the ignition loss was 44.56%. These values suggest that Equation (1) holds for oyster shells, which are primarily composed of calcium carbonate [18].
Calcium carbonate weight ratio (%) = CaCO_3_ molecular weight/CaO molecular weight × CaO weight ratio (%) = 100/56 × 53.66 = 95.82%, There, CO_2_ = approximately 44%(1)

#### 2.2.2. Ordinary Portland Cement

Ordinary Portland cement presented in Section 3.1 of KS L 5201 Portland cement was used. Its physical and chemical properties met the relevant standards [19].

### 2.3. Mix Design

Table 2 shows the mix design for mortar preparation. As mentioned above, the experiment was performed in accordance with KS L ISO 679 (methods of testing cements-determination of strength). The mass ratio of 1:5 (existing mass ratio 1:3) was used by referring to the mix conditions in Section 6.1 of the corresponding standard. This mix design was selected to examine the fire-resistant performance for different particle sizes and OSA contents [15].

## 3. Experiment Results and Analysis

### 3.1. OSA

#### 3.1.1. Density and Water Absorption

Table 3 lists the density and water absorption of the OSA classified according to the experimental conditions. OSA0.6U with the smallest particle size shows the lowest dry density (1.78 g/cm^3^), and OSA2.5 shows the highest density (2.21 g/cm^3^). These observations agree with previous studies and show that OSA is a lightweight material, considering that the generally known density of calcium carbonate is 2.93 g/cm^3^. At the same time, the density was measured in the absolutely dry state; thus, the measured values may be slightly lower than the surface dry density because of the micropores inside the oyster shells [5,10]. Water absorption is the highest (18.93%) for OSA0.6U, which has the smallest particle size and lowest density, and gradually decreases with increasing particle size, showing the lowest value (8.87%) for OSA2.5, which has the largest particle size. As shown in Figure 5, water absorption decreases with increasing density. The water absorption increases with decreasing OSA particle size because of the increase in specific surface area [20]. The high water absorption of OSA (meaning a large amount of micropores) suggests its good thermal insulating properties; however, the pores may also affect strength.

#### 3.1.2. Unit Volume Weight and Solid Volume Percentage

As shown in Table 4, the unit volume weight of OSA increases with particle size from 0.74 kg/m^3^ for OSA0.6U to 0.86 kg/m^3^ for OSA2.5. The solid volume percentage of OSA ranges from 38.9 to 41.6%, which is lower than the solid volume percentage of typical fine aggregate (55–70%) and crushed coarse aggregate (57–60%). This was attributed to the internal pores in OSA. This relationship is graphically expressed in Figure 6. OSA0.6U shows a considerably higher solid volume percentage than other samples because it is the finest powder with particle size of 0.6 mm or less. Therefore, the solid volume percentage of typical OSA was judged to be in the 37–40% range.

#### 3.1.3. Crystal Structure Analysis

Figure 7 shows the results of X-ray diffraction (XRD) analysis, which is commonly used to analyze crystal structure. The peaks in the XRD spectrum for OSA were all attributed to CaCO_3_. In the case of cement, alite (3CaO-SiO_2_) and belite (2CaO-SiO_2_) showed the most prominent peaks. According to a study by Seung-hyo Lee et al., oyster shells contain calcium carbonate in the forms of thermodynamically stable calcite and metastable aragonite [21]. Shellfish generally contain aragonite while calcium carbonate in oyster shells is close to the calcite structure. These structures are compared in Figure 8 [22]. The comparison of structural forms showed that the dissolution of aragonite is faster because its thermodynamic reaction is more unstable than that of calcite. According to the results of previous studies, calcite is generated through the phase transition of aragonite and has the same structure as limestone. However, during heat treatment, both calcite and aragonite decompose into CaO (as the formed CO_2_ escapes) with the same structure [21,23,24].

#### 3.1.4. TGA

As shown in Figure 9a, in OSA0.6U, weight loss starts at 630 °C and ends at 803 °C. In OSA0.6, weight loss starts at a slightly higher temperature of 651 °C and stabilizes at 815 °C. In OSA1.2, the reaction occurs at 687 °C, and the equilibrium is reached at 849 °C. Finally, for OSA2.5, with the largest particle size, weight loss occurs at 688 °C, and the equilibrium is reached at 878 °C. The pyrolysis temperature of typical calcium carbonate is 800 °C or higher. The experimental results, however, show that the decarboxylation reaction in the OSAs with smaller particle sizes occurs earlier than in those with larger particle sizes. As shown in Figure 9a–d, the rate of weight loss decreases with increasing particle size at the same temperature. The slope of the linear trend line fitted to the actual graph is the highest for OSA0.6U (−0.0518) and the lowest for OSA2.5 (−0.0412).

These results show that the actual weight loss and decarboxylation reaction are more rapid in OSAs with smaller particle sizes. This tendency was attributed to the larger heat contact area in OSAs with smaller particle sizes. Based on these experiment results, OSAs with different particle sizes are expected to show different fire resistances even under the same fire conditions.

### 3.2. Mortar

#### 3.2.1. Porosity

The porosity of the mortar specimens was measured using MIP. Table 5 lists the measured porosity and porosity area for mortars containing OSA with different particle sizes. O0.6U-M exhibits the largest porosity area per unit weight (26.3 m^2^/g), whereas O2.5~5.0-M, with the largest particle size, shows the smallest porosity area (20.1 m^2^/g). Similarly, O0.6U-M shows the highest porosity (51.4%) and O2.5~5.0-M the lowest (43.4%), indicating that the inclusion of OSA created internal pores.

As shown in Figure 10a–d, the size of the pores generated in the mortar samples was mostly within the 0.01–1 µm. In the case of O0.6U-M, with the smallest particle size, a peak is observed at 0.1 µm, indicating that a large number of pores with sizes in the vicinity of this value are present in the sample. With increasing OSA particle size, the area of the peak section corresponding to the median value of the 0.1–1 µm range gradually decreases. This tendency was attributed to the effect of the OSA flat grain shape on the cohesion of water, but additional investigations are required [12,25,26]. The obtained results show that when OSA with small particle size is used, relatively small micropores are formed, and the porosity and porosity area are widely distributed. In addition, it was noted that the use of a greater amount of water to meet the flow standard increased porosity because of the difference in water absorption of OSAs with different particle sizes. This finding agrees with the relationship between the water-to-cement ratio and porosity reported in the literature [27].

#### 3.2.2. Back-Side Temperatures Obtained in the Heating Test

After heating for 2 h according to KS F 2257-1, the back-side temperatures of mortar boards were measured, and the results are Figure 11. The back-side temperature of the board containing O0.6U-M was 178.3 °C, O0.6~1.2-M—218.0 °C, and O1.2~2.5-M—237.7 °C. The board made with O2.5~5.0-M showed 273.2 °C, which is more than 90 °C higher than the back-side temperature of the board with OSA0.6U. These results agree with the results of previous studies. Based on the TGA results, such difference was attributed to the greater heat transfer delay caused by higher porosity, porosity area, and specific surface area in OSAs with small particle sizes [12,25,26].

When combined, the TGA and back-side temperature results show that OSAs with smaller particle sizes exhibited greater heat transfer delay because they more rapidly reached the pyrolysis temperature, allowing fast decarboxylation of calcium carbonate, which thereby increased the micropore area.

## 4. Conclusions

In this study, oyster shells were ground into fine aggregate and classified into different particle sizes. The properties of OSAs and mortars containing OSA were investigated. The following conclusions could be drawn.

(1)The TGA results showed that the decarboxylation reaction occurred at 630 °C for OSA0.6U, which had the smallest particle size, and at 688 °C for OSA2.5, which had the largest particle size, indicating that the temperature at which decarboxylation reaction started depended on the OSA particle size. This effect was attributed to the increase in endothermic reaction rate with decreasing particle size because of the increase in specific surface area, i.e., the heat-receiving area.(2)The TGA results combined with the heating test results suggested that CO_2_ would be generated at different temperatures in boards containing OSAs with different particle sizes because of the differences in the endothermic reaction temperature. Therefore, the differences in the back-side temperatures of the boards were attributed to the differences in the insulation effect caused by the thermal decomposition of calcium carbonate and the subsequent generation of micropores and the fire-extinguishing effect caused by the generation of CO_2_.(3)The porosity area and porosity of the OSA-containing mortar increased with decreasing particle size. This was attributed to the effect of the micropores in the OSA and the effect of greater water usage to satisfy the flow standard because of the increase in water absorption with decreasing particle size. The experimental results showed that the porosity and porosity area were proportional and that the back-side temperature decreased with increasing porosity and porosity area.(4)As with the conclusion of the existing literature, high-performance concrete for strength is not always a good decision. Durability and strength of concrete require a low porosity. The low porosity supports all characteristics of concrete with the exception of fire resistance. The lesser the porosity, the higher the risk of spalling, having a final impact on the safety of the infrastructure users [28]. Therefore, the relationship between concrete strength and fire resistance performance is reviewed, and an optimal mixture application study is required as necessary.(5)In a follow-up study, the relationship between the insulation effect and particle size should be further examined by comparing the pore distributions before and after heating. Based on the experimental results, it was concluded that the use of OSA with small particle size is optimal for increasing the fire resistance of mortar boards.

## Figures and Tables

**Figure 1 materials-15-07301-f001:**
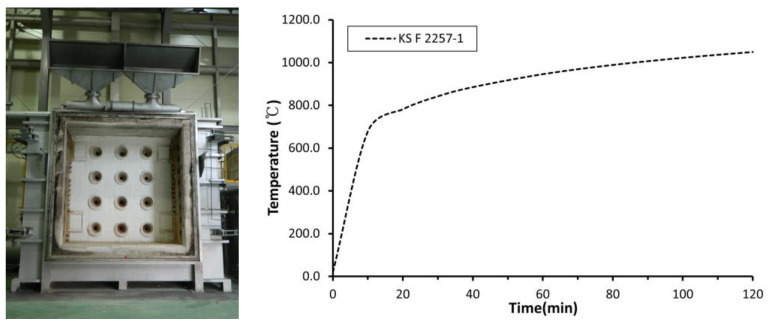
Vertical furnace and heating temperature according to the heating conditions of KS F 2257-1.

**Figure 2 materials-15-07301-f002:**
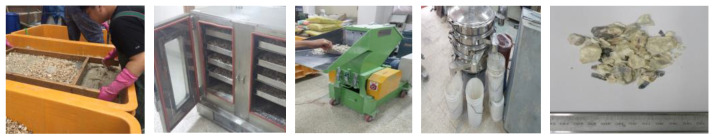
Conversion of oyster shells into OSA (washing, drying, grind in cutter mill, and classification in screen).

**Figure 3 materials-15-07301-f003:**
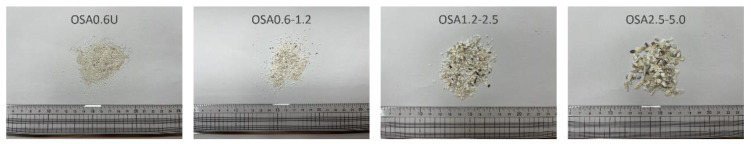
The oyster shell aggregate which was classified according to the particle.

**Figure 4 materials-15-07301-f004:**
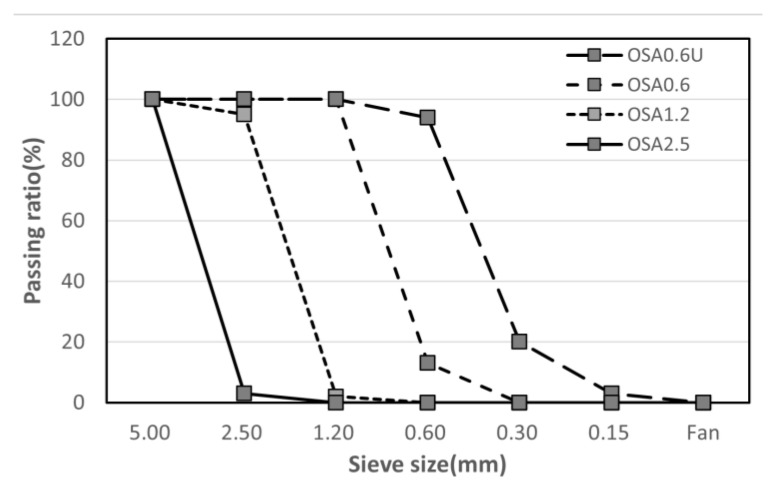
Fineness modulus of oyster shell aggregate with classified.

**Figure 5 materials-15-07301-f005:**
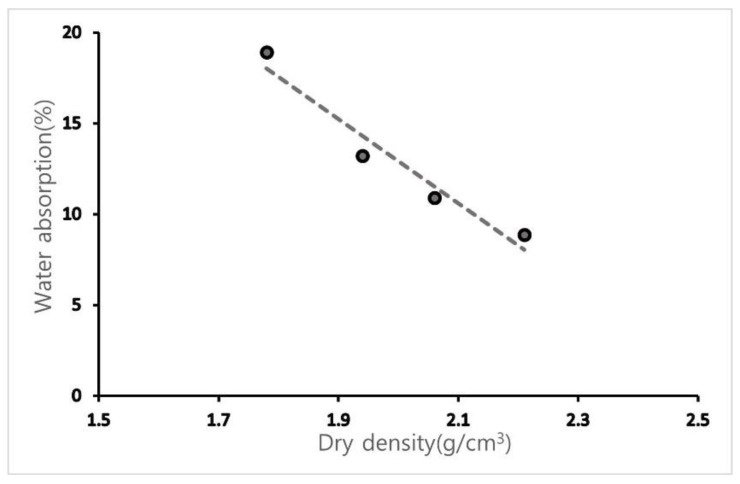
Dry density–water absorption of the oyster shells.

**Figure 6 materials-15-07301-f006:**
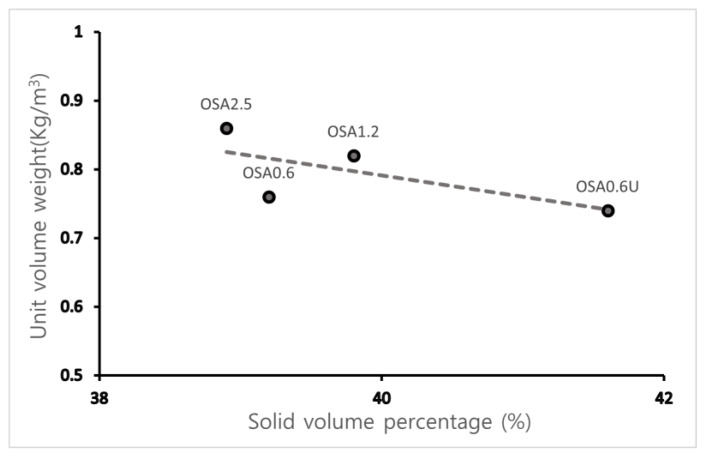
Solid volume percentage–unit volume weight of the oyster shells.

**Figure 7 materials-15-07301-f007:**
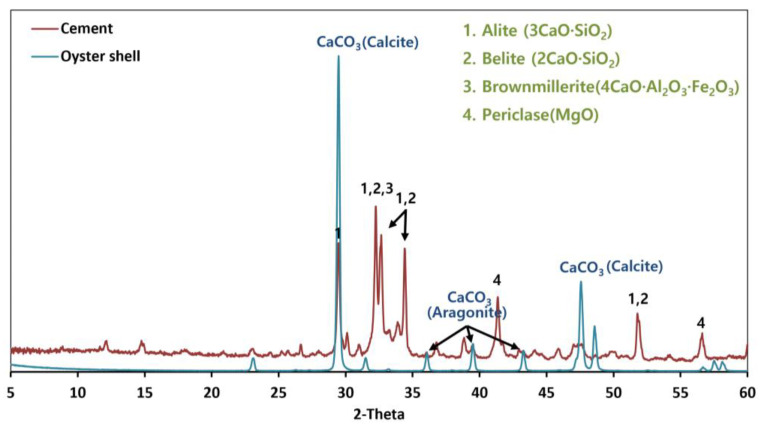
XRD of cement and the oyster shell aggregate.

**Figure 8 materials-15-07301-f008:**
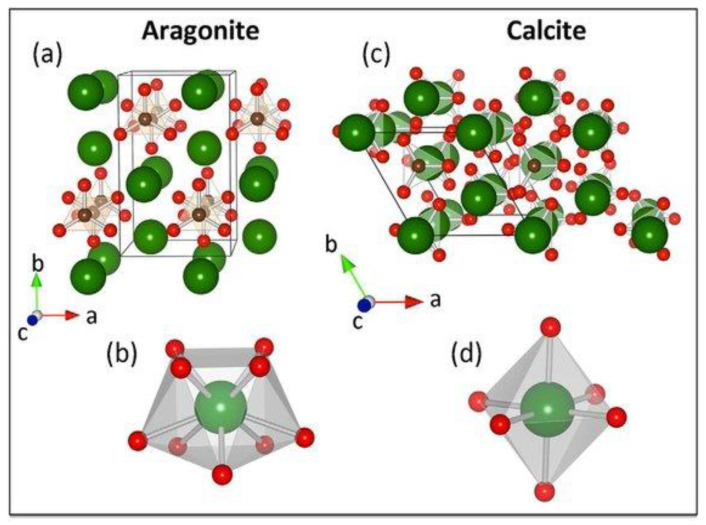
Structures of aragonite and calcite: (**a**) Aragonite; (**b**) Aragonite has a crystal structure; (**c**) In contrast, the crystal structure of calcite; (**d**) Calcium is octahedrally coordinated by O with site symmetry 32.

**Figure 9 materials-15-07301-f009:**
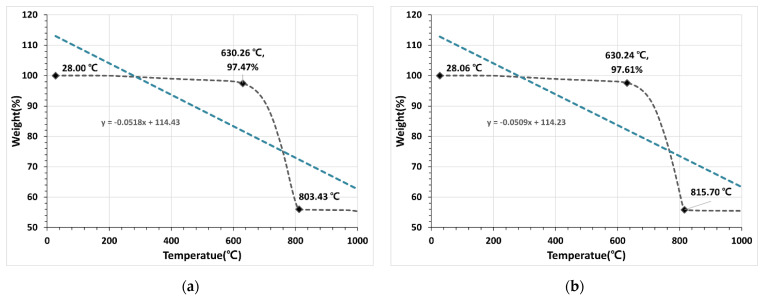
TGA measurement results: (**a**) TGA of OSA0.6U; (**b**) TGA of OSA0.6; (**c**) TGA of OSA1.2; (**d**) TGA of OSA2.5.

**Figure 10 materials-15-07301-f010:**
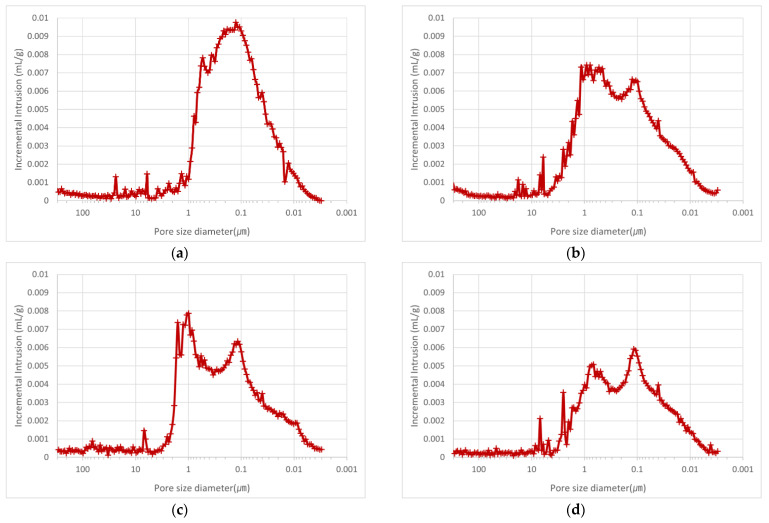
MIP measurement results: (**a**) MIP of O0.6U-M; (**b**) MIP of O0.6~1.2-M; (**c**) MIP of O1.2~2.5-M; (**d**) MIP of O2.5~2.5-M.

**Figure 11 materials-15-07301-f011:**
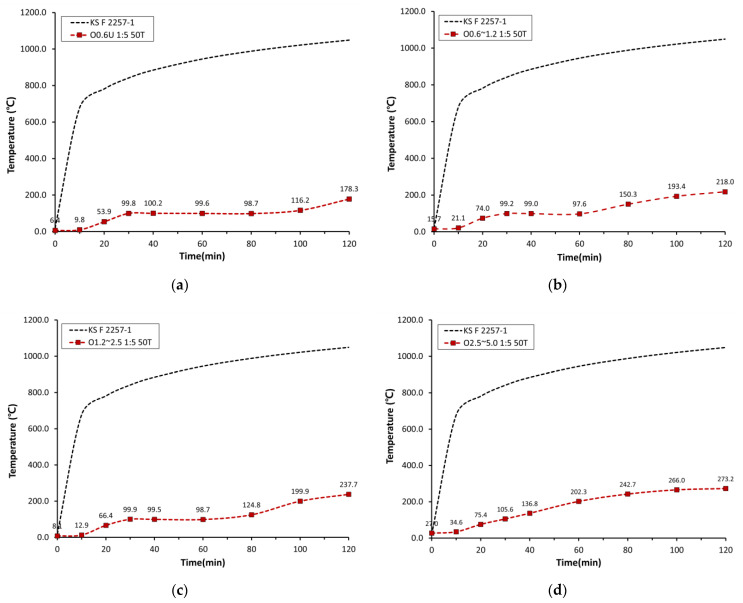
Back-side temperature results: (**a**) O0.6U M (mass ratio 1:5, 50T); (**b**) O0.6~1.2 M (mass ratio 1:5, 50T); (**c**) O1.2~2.5 M (mass ratio 1:5, 50T); (**d**) O2.5~5.0 M (mass ratio 1:5, 50T).

**Table 1 materials-15-07301-t001:** XRF analysis of oyster shell (%).

	SiO_2_	Al_2_O_3_	Fe_2_O_3_	CaO	MgO	K_2_O	Na_2_O	TiO_2_	MnO	P_2_O_5_	Ig-loss *
Oyster shell	0.45	0.12	0.06	53.66	0.26	0.06	0.55	<0.01	0.01	0.16	44.56

* Ig-loss: ignition loss.

**Table 2 materials-15-07301-t002:** Mix design.

ID	Cement (g)	W/C	Aggregate (g)			
			OSA0.6U	OSA0.6	OSA1.2	OSA2.5
O0.6U-M	324.7	0.5	1090.48			
O0.6~1.2-M				1188.50		
O1.2~2.5-M					1262.01	
O2.5~5.0-M						1353.91

**Table 3 materials-15-07301-t003:** Density and water absorption.

ID	Dry Density (g/cm^3^)	Water Absorption (%)
OSA0.6U	1.78	18.93
OSA0.6	1.94	13.22
OSA1.2	2.06	10.89
OSA2.5	2.21	8.87

**Table 4 materials-15-07301-t004:** Unit volume weight and solid volume percentage.

ID	Unit Volume Weight (kg/m^3^)	Solid Volume Percentage (%)
OSA0.6U	0.74	41.6
OSA0.6	0.76	39.2
OSA1.2	0.82	39.8
OSA2.5	0.86	38.9

**Table 5 materials-15-07301-t005:** Porosity and porosity areas of mortar containing OSA with different particle sizes.

Measurement Target	Porosity (%)	Porosity Area (m^2^/g)
O0.6U-M	51.4	26.3
O0.6~1.2-M	50.5	24.4
O1.2~2.5-M	47.1	23.7
O2.5~5.0-M	43.4	20.1

## Data Availability

The data presented in this study are available on request from the corresponding author.

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
