# Peer review of "Characteristics of Mortar Containing Oyster Shell as Fine Aggregate"

_materials, 2022, doi:10.3390/ma15207301_

Round 1

Reviewer 1 Report

it must be made some modifications and some references must be added.

Author Response

Thank you for your careful examination of the paper.

I will do my best to answer the possible parts of your comments.

Reviewer 2 Report

For all the mortar specimens, the flowability is fixed. In this case, the contents of aggregates are not similar for different particle size. How the authors make sure that the properties of mortar are induced by different particle size or by different aggregate volume fraction?

Table 2: The mortar designation is denoted as the same name of the aggregates, which is very confusing. Readers cannot easily recognize what you are referring mortar or just aggregate in the main text. Please change the name of mortar.

Line 161: In addition to internal pores, the particle shape is also responsible for the low solid volume fraction. Do the authors have the typical morphology of OSA?

Line 167: Generally, the solid volume fraction is calculated by the unit volume weight. So, what is the meaning of relating these two parameters?

Line 172: Please cite reference.

Line 187: As seen from Fig. 7, the point of weight reduction is about 630 ℃ for all the aggregates. This means there is no difference for the decarboxylation reaction for the OSA with different particle size. Your conclusion is wrong, based on the presented results.

Figure 7: What does the blue dash line indicate?

Line 216: you stated “OSA flat grain shape”. Please provide the micro images of OSA particles.

Line 263: The water usage in this work is the same for the four mixes. This statement is not correct. Besides, why the authors did not mention the difference of aggregate volume?

Author Response

(The authors gave the same response as above.)

Round 2

Reviewer 1 Report

Please carefully check the last review comments and revise carefully. In particular, suggested references need to be added.

Looking forward to your revision.

Author Response

(The authors gave the same response as above.)
